# Metagenomics Response of Anaerobic Ammonium Oxidation (anammox) Bacteria to Bio-Refractory Humic Substances in Wastewater

**Yabing Meng** [1,2] **, Li-Nan Huang** [3] **and Fangang Meng** [1,2,*]

[1] School of Environmental Science and Engineering, Sun Yat-sen University, Guangzhou 510275, China; mengyb@mail2.sysu.edu.cn

[2] Guangdong Provincial Key Laboratory of Environmental Pollution Control and Remediation Technology, Sun Yat-sen University, Guangzhou 510275, China

[3] School of Life Sciences, Sun Yat-sen University, Guangzhou 510275, China; 13602455700@126.com

[*] Correspondence: mengfg@mail.sysu.edu.cn

**Abstract:** Anammox-based processes have been widely applied for the treatment of wastewater (e.g., wastewater irrigation systems and constructed wetland) which consists of bio-refractory humic substances. Nonetheless, the impacts of bio-refractory humic substances on anammox consortia are rarely reported. In the present study, three identical lab-scale anammox reactors (i.e., HS0, HS1 and HS10), two of which were dosed with humic substances at 1 and 10 mg·L$^{-1}$, respectively, were operated for nearly one year. The long-term operation of the reactors showed that the presence of humic substances in influent had no significant influence on nitrogen removal rates. Despite this, comparative metagenomics showed changes in anammox microbiota structure during the exposure to humic substance; e.g., the relative abundance of *Candidatus* Kuenenia was lower in HS10 (18.5%) than that in HS0 (22.8%) and HS1 (21.7%). More specifically, a lower level of humic substances (1 mg·L$^{-1}$) in influent led to an increase of genes responsible for signal transduction, likely due to the role of humic substances as electron shuttles. In contrast, a high level of humic substances (10 mg·L$^{-1}$) resulted in a slight decrease of functional genes associated with anammox metabolism. This may partially be due to the biodegradation of the humic substances. In addition, the lower dosage of humic substances (1 mg·L$^{-1}$) also stimulated the abundance of *hzs* and *hdh*, which encode two important enzymes in anammox reaction. Overall, this study indicated that the anammox system could work stably over a long period under humic substances, and that the process was feasible for leachate treatment.

**Keywords:** anammox; functional gene; humic substances; *Candidatus* Kuenenia

## 1. Introduction

The distribution of anammox bacteria has been proven to be ubiquitous in various natural environments, such as agricultural soil, inland lakes and marine sediments, accounting for 50%, 4–37% and 9–40% of nitrogen loss, respectively [1]. Anammox was first observed in natural environments in the Baltic–North Sea transition sediments [2], and anammox-related processes have been widely used for wastewater treatment (especially ammonium-rich wastewater) and demonstrated satisfactory performance in nitrogen removal [3]. Compared with the traditional nitrification–denitrification process, anaerobic ammonium oxidation has unique advantages of no external organic carbon usage, the consumption of 62.5% less oxygen and having lower biomass yields [4]. However, due to the low proliferation rates and the high sensitivity of functional bacteria [5], the application of the anammox processes is limited by the complexity of real wastewater (e.g., fluctuations in both compositions

and loads) [6]. According to Obarska-Pempkowiak et al. [7], the concentration of humic substances (HS) in the effluent from wastewater treatment plants is 2.8 to 3.3 mg·L$^{-1}$, while that of constructed wetlands contained 4.2–4.9 mg·L$^{-1}$ of HS. Previously, a number of studies have shown that the organic compounds (e.g., acetate [8], methanol [9] and HS [10]) in wastewater can impact or even inhibit the anammox bacteria. However, there are rare attempts to report the influence of such bio-refractory organics on anammox processes.

HS is a kind of heterogeneous and macromolecular organic matter, which can be derived from organic materials such as soil humus, terrestrial and aquatic plants, animal debris and biomass [11]. In landfill leachate, for instance, HS accounted for more than 60% of total organic carbon [12]. In addition to the landfill leachate, HS is also ubiquitous in natural ecosystems, such as marine sediments and soils, where high anammox activity can be observed [13,14]. Although HS is thought to be inactive in anoxic environments, it can be involved in the microbial reactions in some cases [15]. HS can affect the activity of certain microorganisms by impacting the enzyme systems [16]. Additionally, the degradation byproducts of the HS, such as phenolic compounds, have been found to affect aquatic organisms [17]. In fact, phenolic and quinone-containing compounds can play a role as electron shuttles [18,19]. It must be noted that microbes can also benefit from HS; e.g., from the stimulation of metabolic processes of cells, enhancement of enzyme synthesis, and by acting as electron transfer catalysts during cell respiration [20,21]. Although HS is a type of non-toxic and bio-refractory compound, it can potentially impact the selection, design and operation of anammox-related processes [22]. To this end, it is of high importance to reveal how the presence of HS in wastewater streams influence anammox microbiota, particularly the functional genes of anammox bacteria.

Currently, it is almost impossible to obtain a pure culture of anammox bacteria owing to their unique lifestyle and nutritional requirements [23]. Metagenomics is a powerful approach to capture an all-inclusive picture of microbiota including both cultivatable and uncultivable microorganisms [24,25]. Recently, the draft genome sequences of anammox bacteria affiliated with 'Candidatus Kuenenia' [24], 'Candidatus Jettenia' [26], 'Candidatus Brocadia' [27] and 'Candidatus Scalindua' [28] have been determined, which would encourage the better understanding of the underlying biochemical mechanisms of anammox reaction. Based on the metagenome, the central metabolic pathways of anammox bacteria have been predicted or resolved [29]. Three successive coupled reactions with two intermediates (NO and hydrazine) and three key enzymes (nitrite reductase (*nir*), hydrazine synthase (*hzs*), hydrazine dehydrogenase (*hdh*)) were found to be responsible for the anammox metabolism [29]. Furthermore, the metagenomic study of anammox biofilms also suggested that most of the organisms were capable of nitrate respiration via partial denitrification, which possibly completed a nitrite loop within an anaerobic community by reducing nitrate to nitrite [30]. Therefore, a comprehensive genomic data set for the HS-exposed anammox bacteria would be an important asset to understand the role of bio-refractory HS in regulating the nitrogen cycle.

The major objective in the present study was to investigate the potential impacts of HS on the performance and functional structure of microbiota in anammox reactors. The influence of HS dosage on the long-term performance of nitrogen removal in the anammox reactors was regularly monitored. Comparative metagenomics deep-sequencing analysis was conducted to explore changes in the microbial community structure and functional profile of the anammox consortia. This study could give some clues for dealing with the optimization of anammox-related processes in wastewater treatment and especially the understanding of functional information of anammox bacteria in HS-containing waters.

## 2. Materials and Methods

### 2.1. Reactor Operation

Three identical lab-scale bioreactors, each with an effective working volume of 5 L, were operated under the same operating parameters for approximately one year (Figure S1). Non-woven fabrics with a pore size of 1 μm were used as growth medium [31] and the temperature of the reactors was maintained at 33 ± 2 °C. The reactors were strictly shaded with light-proof black clothes to avoid algal growth which could inhibit anammox bacterial activity. The synthetic wastewater used mainly consisted of ammonium and nitrite in the form of $(NH_4)_2SO_4$ (100 mg-N/L) and $NaNO_2$ (132 mg-N/L), respectively [6]. $NaHCO_3$ (420 mg·$L^{-1}$) was added as an inorganic carbon source for anammox bacterial growth. The wastewater also contained trace element solution I (EDTA and $FeSO_4$) and II (EDTA, $ZnSO_4·7H_2O$, $CoCl_2·6H_2O$, $MnCl_2·4H_2O$, $CuSO_4·5H_2O$, $NaMoO_4·2H_2O$, $NiCl_2·6H_2O$, $NaSeO_4·H_2O$ and $H_3BO_4$) for anammox bacteria activity [32]. All inorganic chemicals (purity 99%) used in the experiments were purchased from Damao (Damao Chemie Co., Ltd, Tianjin, China). The initial biomass concentrations in the reactors were 6.2 g of volatile suspended solids (VSS) per liter (based on 5 L reactor volume). The hydraulic retention time (HRT) was set at about 10 h. No additional sludge was discharged from the reactor (except for the sampling); that is, the reactor was operated at a prolonged solid retention time. The influent pH was maintained at 7.5 ± 0.3. Influent water was continuously introduced into the reactors using a peristaltic pump. The three reactors exhibited satisfying nitrogen removal ability after the startup for about 70 days. Then, HS with increasing concentrations was dosed into the feeding wastewater in the following operating days (from the 70th day on).

The HS stock solution was prepared by dissolving humic acids (sodium salt; Aldrich Chemie Co., Ltd, Shanghai, China) in distilled water. In preliminary research (Figure S2), we used 0, 1, 10 and 20 mg·$L^{-1}$ of HS to study the short-term effect of HS on the nitrogen removal of synthetic wastewater. As the short-term effects of 10 and 20 mg·$L^{-1}$ of HS on nitrogen performance are almost equally significant, 10 mg·$L^{-1}$ of HS was selected as the target concentration to study the long-term effects on nitrogen performance. The control reactor was operated without HS-addition (defined as HS0), while the other two reactors were fed with 1 and 10 mg-HS·$L^{-1}$ (defined as HS1 and HS10, respectively). The biological oxygen demand ($BOD_5$)/chemical oxygen demand (COD) ratio of the HS was determined to be smaller than 0.02, indicating that the HS is a pool of recalcitrant organics for microbes [33].

### 2.2. Sample Collection

Three samples were harvested after HS incubation for molecular biological analysis (281 d). To maintain the homogeneity of biofilm samples, biomass was scraped off the non-woven fabrics and centrifugation was performed at 3500× *g* for 5 min at 4 °C; then, the samples were washed with pre-chilled phosphate buffered saline buffer (PBS, pH 7.5) three times. Samples were centrifuged and the supernatant was discarded. After this, the pellets were stored at −80 °C until DNA extraction.

### 2.3. Metagenomic Sequencing and Bioinformatics Analysis

Community genomic DNA of HS0, HS1 and HS10 were extracted using a Powersoil®DNA Isolation kit (MoBio, Carlsbad, NM, USA). The quality and quantity of extracted DNA samples was monitored routinely in a 1% agarose gel and was measured using a Nanodrop®spectrophotometer (ND-1000 UV spectrophotometer, Thermo Fisher Scientific, Wilmington, DE, USA). DNA samples were sequenced using Illumina HiSeq X Ten platform. After quality control, de novo metagenome assembly was performed with the high-quality reads. Each dataset was assembled using an IDBA UD assembler (version 1.1.3, The University of Hong Kong, Hong Kong, China). The minimum and maximum k-mer lengths were set as 35 bp and 115 bp, respectively [34]. Contigs of more than 500 bp in length were used for further analyses. Open reading frames (ORFs) were predicted from contigs by

using MetaGeneMark (version 2.10, Georgia Institute of Technology, Atlanta, GA, USA) with default settings [35]. CD-Hit was used to remove the redundancy sequences, which are totally identical [36]. For the taxonomic annotation of ORFs, the lowest common ancestor algorithm was selected based on the BLAST (Bell Laboratories Layered Space-Time) results by using MEGAN (Model of Emissions of Gases and Aerosols from Nature) [37].

Assembled contigs were searched against the NCBI database for species annotation (BLAST Coverage Ratio > 40%). The Unigene sequences were blasted against the NCBI non-redundant (NR) protein database using DIAMOND (version 0.4.7, University of Tübingen, Tübingen, Germany). The predicted genes sequences were used to search against the Clusters of Orthologous Genes (COG) [38] and the Kyoto Encyclopedia of Genes and Genomes (KEGG) [39] databases using DIAMOND BLASTX with the E-value cutoff of $1e^{-5}$. The abundance of COG and KEGG entries in each sample was represented by the mean coverage (weighted by their coverage) of genes which belong to this entry. A COG entry was selected when it had more than 10 hits in each of the three samples. For KEGG orthology (KO) entries, more than 5 hits in each sample were required. After filtering, only 227 COG or 128 KO (with a relative abundance of >0.1%) were selected; these were then standardized to z-scores before heatmap plotting [40]. For selected genes including ammonia monooxygenase (*amo*), hydroxylamine dehydrogenase (*hao*), nitrate reductase (*nar*), periplasmic nitrate reductase (*nap*), nitrite oxidoreductase (*nxr*), nitric oxide reductase (*nor*), nitrous oxide reductase (*nos*), nitrite reductase (*nrf*), *nir*, *hzs* and *hdh*, their annotation information was manually obtained from the nitrogen metabolism based on the KEGG database. A similarity cut-off of >50% was used for the relative abundance of *hzs* and *hdh* sequences in the three samples. Furthermore, to extract reads matching the ribosomal small subunit for taxonomic classification, the SILVA SSU RefNR99 dataset (version 115, www.arb-silva.de) was used as a reference [41]. Additionally, DNA sequences from this study have been submitted to the NCBI repository under the BioProject record PRJNA397254.

We calculated the relative abundance of every taxon using the following equation [42]:

$$\text{Relative abundance} = \frac{a}{b} \times 100\,\%$$

where *a* represents the number of sequences that were assigned to the taxon, and *b* represents the total number of sequences that were assigned to all the taxa. The relative abundance of a given gene, as well as the KEGG pathway, KEGG subcategory, COG and COG category were also obtained in a similar calculation.

### 2.4. Chemical Analyses

The influent and effluent water quality of the anammox biofilm reactors were monitored thrice a week during the operating time. The concentrations of ammonium, nitrite, total nitrogen, chemical oxygen demand (COD) and biological oxygen demand ($BOD_5$) were determined according to the standard methods [43]. The sludge samples were washed with pre-chilled phosphate buffered saline (PBS) buffer three times for further heme analysis. The *c*-type heme content was determined according to the Berry method [44].

## 3. Results and Discussion

### 3.1. Reactor Operation

During the one-year operation, the nitrogen removal rates (NRR) of $0.59 \pm 0.03$, $0.63 \pm 0.03$ and $0.59 \pm 0.03$ g-N·L$^{-1}$·d$^{-1}$ ($p > 0.05$) were achieved at the HS of 0, 1, 10 mg·L$^{-1}$, respectively, at a nitrogen loading rate (NLR) of $0.79 \pm 0.15$ g-N·L$^{-1}$·d$^{-1}$ (Figure 1a). Robust operation of the anammox systems was observed in spite of incubation with different HS concentrations. This phenomenon was also corroborated by the similar heme content ($0.67 \pm 0.12$, $0.70 \pm 0.12$ and $0.68 \pm 0.12$ μmol·g-VSS$^{-1}$; Figure 1b) among the three reactors, which can indicate the activity of anammox bacteria [45,46].

Furthermore, lower nitrate concentrations were obtained in HS10 ($20.88 \pm 1.58$ mg·L$^{-1}$) than those produced in HS0 ($26.20 \pm 2.02$ mg·L$^{-1}$) and HS1 ($31.45 \pm 2.44$ mg·L$^{-1}$), which was further evidenced by the higher alkalinity in 10 mg-HS/L ($385.82 \pm 5.96$ mg·L$^{-1}$) than in 0 mg-HS/L ($338.86 \pm 4.90$ mg·L$^{-1}$) and 1 mg-HS/L ($348.26 \pm 6.41$ mg·L$^{-1}$). These results indicated a potential denitrification at high HS concentrations, which was consistent with the findings of Dong and Chen [47] and Chatterjee et al. [48]. These authors reported that, after a long-term operation at 10 mg-HS/L, anaerobic respiratory bacteria could decrease the final nitrate concentration and thus promote the denitrification process. Additionally, the effluent pH was $8.1 \pm 0.2$, $8.3 \pm 0.2$ and $8.2 \pm 0.2$ ($p > 0.05$) in response to HS0, HS1 and HS10, respectively, which is the ideal pH for the anammox process [46].

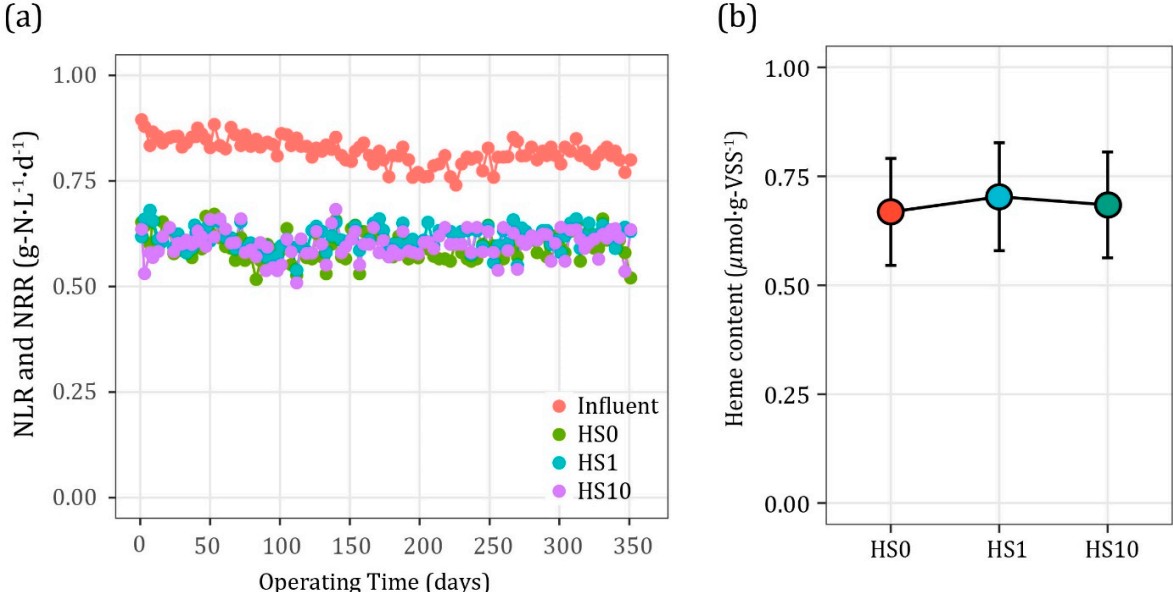

**Figure 1.** Reactor performance of anammox bioreactors and physiological dynamics of anammox bacteria under different humic substance concentrations. (**a**) Nitrogen loading rate (NLR) and nitrogen removal rate (NRR) during 351 days of operation. (**b**) Heme content. HS0, HS1, HS10 indicate 0, 1, 10 mg·L$^{-1}$ humic substances, respectively.

### 3.2. Microbial Community Structure

Relative abundances of taxa were calculated by mapping the high-quality reads back to the assembled contigs and then counting the number of reads assigned to each taxon. Sequences annotated to *Planctomycetes* as the dominant phylum were higher in HS0 (34.71%) and HS1 (33.53%) than that in HS10 (28.70%), implying that the lower relative abundance may have resulted from the fact that microbial populations metabolizing HS were stimulated (Figure 2). The composition of the microbiome was also analyzed using 16S rRNA gene reads retrieved from the metagenomes (Figure S3), showing that HS could decrease the abundance of dominant phylum *Planctomycetes* (43.21%, 37.27% and 37.42% in HS0, HS1 and HS10, respectively). A recent study showed that anammox bacteria could adapt to the phenol environment in industrial coke-oven wastewater sludge [49]. In line with the relative abundance of *Planctomycetes*, *Candidatus* Kuenenia was determined to be the most dominant genus in anammox bacteria, whose relative abundance showed a declining trend under the three treatments (22.84%, 21.68% and 18.51% for HS0, HS1 and HS10, respectively). Guo et al. [50] and Kuenen [51] also found that *Candidatus* Kuenenia was the dominant structuring force of anammox microbiota in lab- or pilot-scale reactors. However, our findings differ from an investigation of aquaculture systems; i.e., positive correlations between organic matter and Kuenenia-like anammox bacteria were found [52].

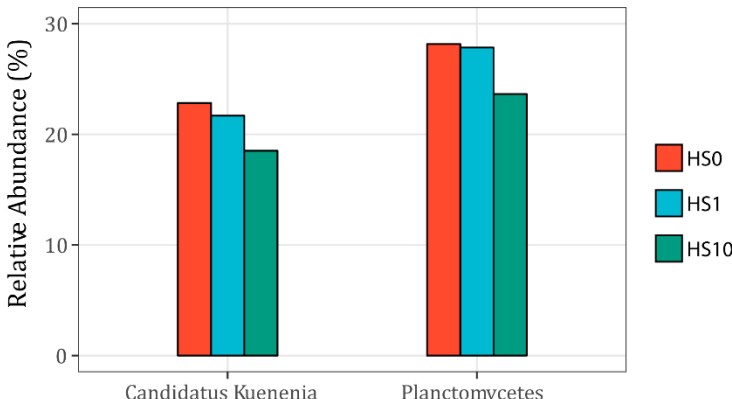

**Figure 2.** Relative abundance of the anammox community under different humic substance concentrations. HS0, HS1 and HS10 indicate 0, 1 and 10 mg·L$^{-1}$ humic substances, respectively.

Furthermore, due to the high similarity of the metagenomes between "*Candidatus* Kuenenia" and "*Candidatus* Brocadia" [53], it was not surprising that they were simultaneously detected in many ecosystems. In the present study, the genus *Candidatus* Brocadia was slightly higher during the exposure to low HS (HS1; 2.17%), compared with HS0 (0.90%) and HS10 (1.39%) (Table S1). Notably, *Candidatus* Brocadia has a more versatile metabolism and better adaptability than other anammox bacteria [54], suggesting a good adaptation of *Candidatus* Brocadia to resource-abundant environments. Recently, *Candidatus* Brocadia has been proved to be the dominant anammox bacteria in the landfill leachate treatment bioreactors [10], suggesting this bacterium has broad ecophysiology and competitiveness, which justifies its wide adaptation in different engineering systems. Overall, although there was no significant difference in the NRR of the three reactors ($p > 0.05$), the dominant phylum and genus responsible for anammox reaction were decreased in HS10. Recent studies have shown that the presence of organic compounds in water could lead to the dominance of Brocadia lineage [55,56]. On the other hand, organic matter, particularly biodegradable matter, can inhibit the activity of potential anammox bacteria [6]. It has also been documented that the presence of high concentrations of humic matter inhibited the activity of microbes [25]. Beyond a certain concentration, the presence of excess HS in a reactor or system often gives rise to a reduction in microbial activity, which is commonly due to a restricted diffusion of nutrients and oxygen to the cells [57]. In this present study, we can speculate that *Candidatus* Kuenenia and *Candidatus* Brocadia could adapt well to the low HS concentration (1 mg·L$^{-1}$).

Furthermore, archaea occurred in our reactor microhabitat (0.47%, 0.44% and 0.40% in response to HS0, HS1 and HS10; similarly hereafter), based on the analysis of the metagenome samples. This result was generally consistent with a previous metagenomics analysis of anammox biofilms, where archaea also showed a very low abundance [50], which is similar to Anielak et al. [58], who interpreted that fulvic acids involved in HS were resistant to biodegradation but were susceptible to archaea. Likewise, the phylum Cyanobacteria showed a declining trend in the three reactors (i.e., 2.02%, 1.90% and 1.70%), which was positively correlated with anammox bacterial abundance in the present study. This is in agreement with previous observations focusing on reservoir ecosystems [59]. Evidently, anammox bacteria often coexist with ammonium-oxidizing bacteria (AOB) and nitrite-oxidizing bacteria (NOB) under oxygen-limited conditions [60]. AOB would provide nitrite for anammox bacteria and NOB, while NOB would compete with anammox bacteria for nitrite and with AOB for oxygen [50]. However, in the present study, AOB and NOB across the three samples showed extremely low abundance in all communities (<1%) (Figure S4; Table S2), indicating that the contribution of nitrification to nitrogen conversion in our reactors was ignorable. Moreover, *Acidovorax* ebreus, which can reduce nitrate to nitrogen [61], was more abundant in HS10 (0.061%) than in HS0 (0.016%) and HS (0.005%), suggesting that heterotrophic denitrobacteria are more active and responsive to organic-rich environments. Overall, the results in this present study indicated that AOB, NOB and

denitrobacteria, as a result of their low abundances, may not be the potential contributors to nitrogen conversion in the anammox community of the three reactors.

### 3.3. Enrichment of Biological Functions

We obtained 17.4 million reads from the three samples. After assembly, 23,464,834 reads were assembled into 68,869 contigs (HS0), 22,479,031 reads into 72,451 contigs (HS1) and 22,593,532 reads into 84,185 contigs (HS10). Additionally, metagenome sequences obtained 206,324, 202,417 and 230,257 genes from HS0, HS1 and HS10 communities, respectively (Table 1). These genes were functionally annotated using the COG and KEGG databases (Figure 3). We found a potential difference in metabolic capabilities between HS1 and the other two concentrations (HS0 and HS10) (Figure 2, Figures S5 and S6; Tables S3 and S4). Metagenome sequences were assigned to 23 COG categories, in which 7 functional categories were significantly enriched in HS1 (Figure S5; Table S3). In particular in HS1 communities, a large proportion of genes were assigned to general function prediction only (COG category [R]) or function unknown (COG category [S]), showing large pools of potential unknown functional genes in low HS-containing reactors [62]. Of note, the genes associated with secondary metabolite biosynthesis, transport and catabolism (COG category [Q]) were much higher in HS1 (4.40%) than in HS0 (1.07%) and HS10 (1.01%), indicating that the secondary metabolites production was more active in the HS1 reactor. Furthermore, a large amount of genes involved in signal transduction mechanisms (COG category [T]) (5.43%, 7.72% and 5.33%), intracellular trafficking, secretion, and vesicular transport (COG category [U]) (1.55%, 4.73% and 1.53%) and defense mechanisms (COG category [V]) (3.44%, 6.19% and 3.43%) were more abundant in the HS1 reactor relative to the other two reactors. This implied that some functional proteins can adapt HS exposure via the regulation of signaling cascade mechanisms and defense mechanisms. However, the genes associated with basal metabolisms, such as energy metabolism (COG category [C]) and amino acid transport and metabolism (COG category [E]), were low in HS1, suggesting the deficiency of substance and energy metabolism in the HS1 environment. These results suggested complex interactions of anammox bacteria with HS in the reactors. In general, COG functional categories revealed that anammox bacteria have divergent distributions of functional categories in the HS environments, indicating that they had different responsibilities for cellular functions.

Furthermore, the major portion of the contigs found in the metabolism class were dedicated to carbohydrate metabolism (11.05%, 11.25% and 17.73%) (Figure S6; Table S4), followed by amino acid metabolism (10.44%, 11.43% and 17.40%), nucleotide metabolism (4.24%, 4.36% and 6.96%) and lipid metabolism (3.66%, 3.85% and 5.83%). Data presented here showed that the genes involved in the metabolism class had much higher abundance in HS10 relative to those in HS0 and HS1 (Figure S3, Table S4). The metabolism class is one of the most crucial processes for the survival of most organisms. Thus, we conjecture that bacteria can more actively transcribe certain genes under a specific condition, thus strengthening their metabolic levels and possibly resulting in higher cellular activity. This could explain the fact that even at decreased bacterial abundance, the anammox bioreactor could still maintain high nitrogen removal rates.

**Table 1.** Sequencing, assembly and annotation statistics.

| Sample | High-Quality Reads (M) | Assembled Reads (%) | Contigs > 300 bp | Contigs N50 (bp) | ORFs Total | Annotated ORFs by COG (%) | Annotated ORFs by KEGG (%) |
|---|---|---|---|---|---|---|---|
| HS0 | 1.80 | 83.23 | 68,869 | 7752 | 206,324 | 61.09 | 70.56 |
| HS1 | 1.74 | 83.23 | 72,451 | 5105 | 202,417 | 61.50 | 70.55 |
| HS10 | 1.78 | 81.26 | 84,185 | 4518 | 230,257 | 61.76 | 71.08 |

Note: HS0, HS1, HS10 indicate 0, 1, 10 mg·L$^{-1}$ humic substances, respectively. ORF: open reading frames; COG: Clusters of Orthologous Genes; KEGG: Kyoto Encyclopedia of Genes and Genomes.

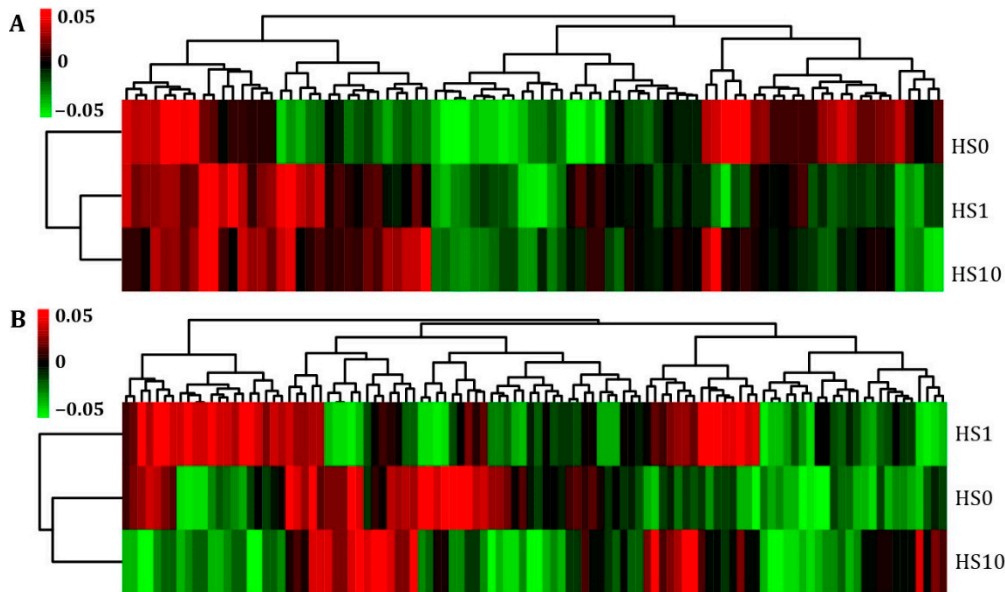

**Figure 3.** Clustering of the functional gene abundances under different humic substance conditions based on the functional annotation of the genes by (**A**) COG and (**B**) KEGG pathways. HS0, HS1 and HS10 indicate 0, 1 and 10 mg·L$^{-1}$ humic substances, respectively.

### 3.4. Functions in Nitrogen Metabolism

In order to further reveal the molecular mechanisms driving the functional diversification of the anammox microbiota, taxonomic analyses based on key genes associated with nitrogen metabolism were performed. Clearly, the nitrogen conversion in an anammox reactor was conducted by a diverse range of genes and bacteria (Figure 4). In the present study, the abundances of *hzs* and *hdh*, which are related to anammox bacteria, were promoted upon exposure to low HS (1 mg·L$^{-1}$). A previous study also found that anammox was able to adapt to a high strength of phenol toxicity, without losing its anaerobic ammonium-removing activity [49]. However, the abundance of bacterial *amoA*, which is mostly associated with *Nitrosomonas*, was higher in HS10 than in HS1, indicating that *Nitrosomonas* was strongly promoted by high-HS exposure (10 mg·L$^{-1}$). This result is supported by a previous work [63], which showed that *Nitrosomonas* and *Planctomycetes* co-occurred in landfill leachate treatment bio-processes and *Nitrosomonas* can tolerate high-HS concentration. Despite this, the abundance of AOB in HS10 was still as low as 1%. In addition, *NorB* was mostly assigned to *Proteobacterium* in the three samples (1.50%, 1.13% and 1.72%), which was congruent with the fact that *Proteobacteria* were well adapted to humus content [64].

Besides this, the coexistence of microbes belonging to the phylum *Chloroflexi* is usually reported in anammox bioreactors, which can utilize the cell debris as a carbon source [65]. In particular, Shu et al. [66] found that *nrfA* was the crucial functional gene determining the organotrophic anammox contribution, and *Planctomycetes*, *Proteobacteria* and *Chloroflexi* were the most abundant phyla in the organotrophic anammox system. Consistently, the abundance of *Chloroflexi*, which harbored with *nrfA*, was higher at HS1 than at HS0 and HS10. We should note that there could be a small portion of organics in the HS pool that are utilizable for microbes. Thus, the exposure to a high HS can support the growth or activity of some heterotrophs [67], finally leading to the up-regulation of some functional genes. Furthermore, the conserved microbiota composition obtained by metagenomics analysis suggests overall that the presence of HS in water or wastewater can to a certain extent mediate the nitrogen metabolism pathways, although no significant differences were found for the NRRs of the three reactors. We hypothesized that, although the predominant functional bacterial population decreased at high HS conditions, increased activity in functional microorganisms occurred, aiding in maintaining a stable nitrogen removal capacity.

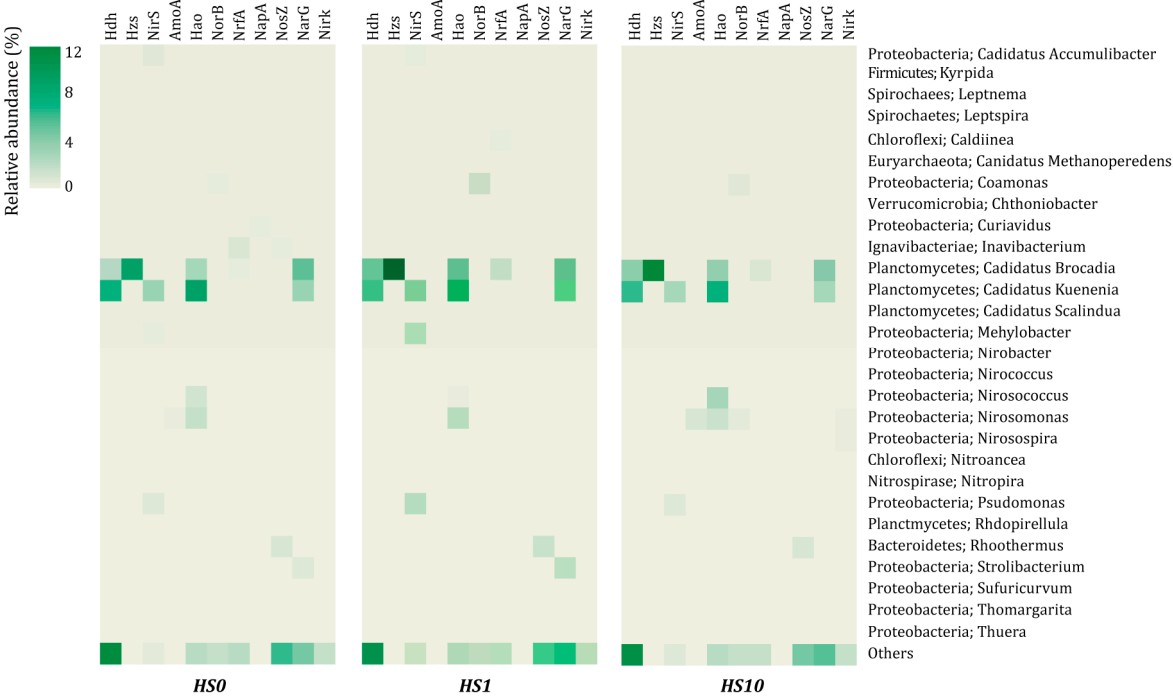

**Figure 4.** Taxonomic origins of the key enzymes in nitrogen metabolism based on metagenome sequences. *amoA*, bacterial ammonia monooxygenase subunit A; *narG*, cytoplasmic nitrate reductase alpha chain; *napA*, periplasmic nitrate reductase precursor; *nirK*, copper nitrite reductase; *nirS*, cytochrome cd1 nitrite reductase; *norB*, nitric oxide reductase subunit B; *nosZ*, nitrous oxide reductase; *nrfA*, ammonia-forming dissimilatory nitrite reductase; *hzs*, hydrazine synthase; *hdh*, hydrazine dehydrogenase. HS0, HS1 and HS10 indicate 0, 1 and 10 mg·L$^{-1}$ humic substances, respectively.

## *3.5. Quantification of Nitrogen-Related Genes*

To gain a better understanding of functional diversity, especially the nitrogen metabolism pathways, an abundance of nitrogen-related genes was present (Figure 5). Nitrite and nitrate resulting from nitrification can be readily used in denitrification, anammox and dissimilatory nitrate reduction to ammonium (DNRA). AOB and NOB encoded the key enzymes for ammonia (*amo*) and nitrite oxidation (*nxr*), respectively. The *amoA*, which is the gene encoding key enzyme of aerobic ammonium oxidation during the first step of nitrification, were much higher in HS10 (0.006%) than those in HS0 (0.002%) and HS1 (0.001%), indicating that *amoA* was stimulated at a high HS level. The second step of nitrification, nitrite oxidation, depends on the presence of two homologous enzymes; i.e., nitrite oxidoreductase (*nxrA*) and nitrate reductase (*narG/napA*). It is interesting to note that the abundance of *nxrA* potentially increased slightly during the exposure to HS stress (0.006%, 0.007% and 0.007%). It is interesting to note that *nirS* only made up 0.056%, 0.059% and 0.058% in HS0, HS1 and HS10, respectively. In addition, humus content had little impact on the *nosZ* gene (0.052%, 0.047% and 0.044%). A previous study showed that straw and biochar barely affect the *nosZ* in a microbial community [68]. Although the reactor performance was not remarkably impacted by the HS concentrations, the relative abundance of key genes involved in anammox bacteria was slightly higher in the low-HS reactor than those in the control and high-HS reactor. During the anammox reaction, nitrite is reduced to nitric oxide (NO) [69], and there is simultaneous condensation to produce hydrazine (N$_2$H$_4$) with ammonium [70]. *Hzs*, as one of the key features of anammox bacteria, catalyzes the synthesis of N$_2$H$_4$. It is interesting to note that the relative abundance of *hzs* was higher in HS1 (0.101%) than in HS0 (0.056%) and HS10 (0.076%). It has been suggested that the presence of HS can cause changes in metabolism, allowing organisms to proliferate on the substrates that they could not use without HS [71]. Furthermore, *hdh*, as another key enzyme that catalyzes the oxidation of the intermediate N$_2$H$_4$ to dinitrogen gas [29], was slightly

stimulated under low-HS conditions (0.133%, 0.161% and 0.137%). This could potentially increase the anammox activity at 1 mg-HS L$^{-1}$. Thus, the relative abundance of functional genes involved in anammox metabolism were the main factors determining nitrogen removal.

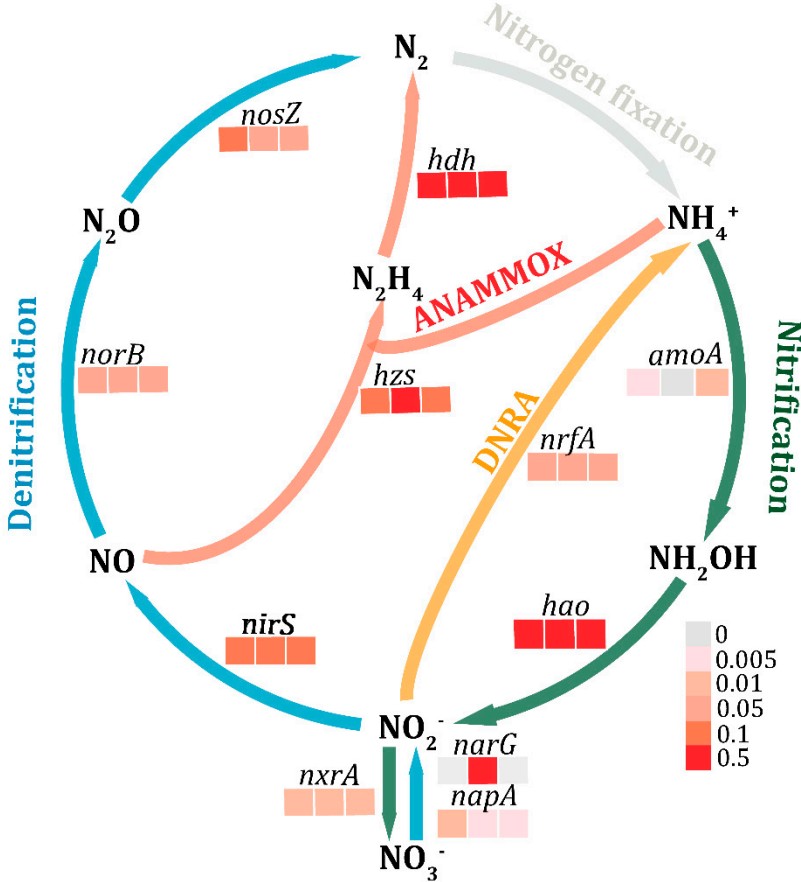

**Figure 5.** Relative abundance of microbiota and functional genes in the anammox bioreactors. *amoA*, bacterial ammonia monooxygenase subunit A; *hzo*, hydroxylamine oxidoreductase; *narG*, cytoplasmic nitrate reductase alpha subunit; *napA*, periplasmic nitrate reductase precursor; *nrfA*, ammonia-forming dissimilatory nitrite reductase; *nxrA*, nitrite oxidoreductase alpha subunit; *nirS*, cytochrome *cd₁* nitrite reductase; *norB*, nitric oxide reductase subunit B; *nosZ*, nitrous oxide reductase. The abundance of the enzymes in each sample is shown in the color bars, and the location in the bar from left to right shows HS0, HS1 and HS10.

In addition, the nitrate reduction is normally catalyzed by the *napA* and *narG* genes [72]. As compared to the *napA* gene (0.005%, 0.004% and 0.004%), the relative abundance of the *narG* gene was remarkably higher in three samples (0.103%, 0.115% and 0.100%). Therefore, the results indicated that *narG* was the main functional gene that influeced the reduction of NO$_3$-N. Furthermore, the relative abundance of dissimilatory nitrogen reduction gene *nrfA,* which encodes the key enzyme of DNRA, increased slightly from 0.030% to 0.043% at HS0 to HS1. Previous studies have observed that some anammox bacteria could participate in "disguised denitrification", also known as DNRA [73]. Furthermore, the average depth of the *narB* (1.12, 1.07 and 1.93) gene, which is involved in the assimilatory nitrate reduction, showed a minor increase under high HS levels (HS10). Kartal et al. [73] found that anammox bacteria could mediate DNRA under ammonium-limited conditions.

The key drivers that shape microbial communities are competition for substrates and collaboration for providing intermediates or the removal of toxic substances. As alluded to above, when we analyzed the nitrogen cycling metabolic potential of the anammox microbiota, it was worth noting that gene redundancy in well-described pathways occurred [74]. Briefly, although the NRR did

not unequivocally reconcile with the relative abundance of anammox bacteria when subjected to different HS concentrations, it seemed that the presence of HS in influent water plays a dual role; i.e., stimulating gene expression of the nitrogen pathway and triggering the metabolism or growth of some heterotrophs. It is worth noting here that the presence of a certain HS can benefit microbes by regulating many physiological and biological functions, rather than the binding and release of nutrients [75]. Furthermore, the HS can serve as a source of additives that can stimulate biological activity and remove toxic inhibitors of the biological processes [75].

*3.6. Implications of This Work*

The anammox process has been proved to be an effective way for nitrogen removal from landfill leachate which contains high-strength ammonium and complex organic substances [76,77]. However, it is common that organic matters become refractory with the aging of the landfills [10]. Organic nitrogen was thought to be related to humic substances (i.e., HS) and was shown to be more recalcitrant for biodegradation [78,79].

Previous studies have mainly focused on the roles of either the microbial community or the microbial interactions in determining the anammox performance, yet the molecular-level characterization of microbial interactions in the anammox performance under HS conditions has not been paid sufficient attention. In this study, therefore, the influence of HS on biological processes and the microbial community was discussed. Our current findings demonstrated that using the anammox process for treating HS-contained wastewater exhibited stable nitrogen removal. However, high-level HS (10 mg·L$^{-1}$) led to a decrease of the abundance of dominant phylum *Planctomycetes* and genus *Candidatus* Kuenenia as well as the functional genes. This phenomenon was most likely attributed to their intrinsic adaptation strategies. Thus, it can be expected that the presence of low HS potentially benefited the efficiency of anammox in wastewater treatment. In addition, future research is needed to explore the response of bacteria activities to HS based on transcriptomics and metabolomics. Overall, our results can not only improve our knowledge of the influence of HS on anammox systems, but are also expected to supply useful information for optimizing the operation of anammox-related systems.

## 4. Conclusions

In the present study, we have investigated the potential influence of bio-refractory HS on the reactor performance and functional structure of microbial communities within anammox bioreactors. The key findings are as follows:

(1) The nitrogen removal performance of the anammox reactors was not significantly impacted by the presence of HS in wastewater;

(2) High-level HS (10 mg·L$^{-1}$) in wastewater led to a decrease of the abundance of phylum *Planctomycetes* and genus *Candidatus* Kuenenia, whereas it led to a slight increase for genus *Candidatus* Brocadia. This suggests that the different adaptability of these two genera to HS;

(3) Low-level HS (1 mg·L$^{-1}$) in wastewater resulted in the increases of signaling genes, whereas high-level HS (10 mg·L$^{-1}$) induced the increase of genes involved in the metabolism of organic matter. The responses of functional genes, which are responsible for nitrogen metabolism (*amoA*, *NorB*, *amo*, *nxr*, *narG/napA*, *nosZ*, *hzs*, *hdh*), to HS were different from each other.

**Supplementary Materials:** The following are available online at http://www.mdpi.com/2073-4441/11/2/365/s1, Figure S1: Schematic diagram of the lab-scale anammox bioreactors, Figure S2: The short-term effects of HS on the nitrogen removal for synthetic wastewater, Figure S3: The community's taxonomic composition calculated from metagenomic data by extracting the marker gene for classification, Figure S4: Abundances of AOB, NOB and anammox microorganisms in the three samples as determined from metagenome sequences, Figure S5: Microbial gene abundances, grouped using the Clusters of Orthologous Genes (COG) hierarchy of functions, Figure S6: Functional categories of Kyoto Encyclopedia of Genes and Genomes (KEGG) category profiles of the three samples, Table S1: Species and abundance of anammox bacteria detected by metagenomic data, Table S2: Abundance comparison of AOB, NOB and anammox based on metagenomic sequencing in three samples, Table S3: Clusters of Orthologous Genes (COG) annotation statistics, Table S4: Kyoto Encyclopedia of Genes and Genomes (KEGG) annotation statistics.

**Author Contributions:** Conceptualization, Y.M. and F.M.; Methodology, L.H.; Software, Y.M.; Validation, F.M. and L.N.; Formal Analysis, Y.M.; Investigation, F.M.; Resources, F.M.; Data Curation, Y.X.; Writing-Original Draft Preparation, Y.M.; Writing-Review & Editing, F.M.; Visualization, Y.M.; Supervision, F.M.

**Funding:** This research was funded by the National Natural Science Foundation of China grant number 51622813 and 21107144 and Science and Technology Planning Project of Guangdong Province, China (No. 2017B020216006 and No. 2015A020215014).

**Conflicts of Interest:** The authors declare no conflict of interest.

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
