# Peer review of "Metagenomics Response of Anaerobic Ammonium Oxidation (anammox) Bacteria to Bio-Refractory Humic Substances in Wastewater"

_water, doi:10.3390/w11020365_

Round 1

Reviewer 1 Report

This paper discusses the meta-genomic response of anaerobic ammonium oxidation bacteria (ANNAMOX) to humic substances in wastewater, providing useful and interesting insights on performance and functional structure of microbial communities in ANNAMOX reactors.

However, there are some major issues which need to be addressed.

Introduction

Authors presents the main objective of their study, but I believe they should also include a paragraph where they explain why ANNAMOX- based processes are important, when they are preferably used and what are their advantages over other remediation techniques with similar Nitrogen removal performances.

Materials and methods

This section needs major improvements. Authors need to explain why they used synthetic wastewater, include references and comment on the implications of such choice (over the use of real wastewater). They need to include more information on the chemicals used (brand, purity etc) as well as on the selection of the non-woven material as growth medium support.

Authors report they used essential trace elements. What are they? Would they be found in natural conditions or would need to be supplemented in real wastewater?

Also, they need to provide an explanation on why the two concentration of HS (1 and 10 mg/L) were selected.

I believe this section would be further improved if authors would provide a schematic representation of the reactors design.

Enrichment of biological functions

Line 271: define what is meant by “satisfying” removal performance.

Conclusions

Authors need to expand this section by providing suggestions and advice, based on their findings, on future research efforts on the topic.

Author Response

Reviewer #1 (Comments for the Author):

This paper discusses the meta-genomic response of anaerobic ammonium oxidation bacteria (ANNAMOX) to humic substances in wastewater, providing useful and interesting insights on performance and functional structure of microbial communities in ANNAMOX reactors.

However, there are some major issues which need to be addressed.

We thank the reviewer for the positive assessment on our manuscript.

# Introduction
Authors presents the main objective of their study, but I believe they should also include a paragraph where they explain why ANNAMOX- based processes are important, when they are preferably used and what are their advantages over other remediation techniques with similar Nitrogen removal performances.

Author Response: Thanks for the comments. Firstly, as presented in the paragraph one, we have already mentioned the importance of ANAMMOX process (Line NO. 31-36). Secondly, as per the reviewer’s suggestion, the usage and the advantages of ANAMMOX have been added in this revised manuscript (Line NO. 35-38).

Line No. 35 (Revised MS):
ANAMMOX was first observed in natural environments in the Baltic-North Sea transition sediments [2], the ANAMMOX-related processes have widely used for wastewater treatment (especially ammonium-rich wastewater) and demonstrated satisfactory performance in nitrogen removal [3].

Line No. 36-38 (Revised MS):
Compared with the traditional nitrification-denitrification process, anaerobic ammonium oxidation has its unique advantages: no external organic carbon usage, consume 62.5% less oxygen and have lower biomass yields [4].

# Materials and methods
This section needs major improvements.

1) Authors need to explain why they used synthetic wastewater, include references and comment on the implications of such choice (over the use of real wastewater).

Author Response: Firstly, the composition of synthetic wastewater is simple which could keep the reactor system stable. Secondly, due to the simple composition, the synthetic wastewater can clearly reflect the changes in nitrogen removal efficiency. In contrast, the real wastewater contains complex substances and variable compositions, which cannot maintain the stability of the reactor performance. Therefore, if the impact factor is changed, we have no way to know which factors cause the changes in the reactor performance. Considering the above reasons, we choose synthetic wastewater as the influent.

    In this revised version, relevant references have been included for the use of the synthetic wastewater (Line NO. 149).

Line No. 149 (Revised MS):
The synthetic wastewater used mainly consisted of ammonium and nitrite in the form of (NH4)2SO4 (100 mg-N/L) and NaNO2 (132 mg-N/L), respectively [6].

2) They need to include more information on the chemicals used (brand, purity etc) as well as on the selection of the non-woven material as growth medium support.

Author Response: Information for the chemicals has been added.

Line No. 101 (Revised MS):
Non-woven fabrics with a pore size of 1 μm were used as growth medium [31] and temperature of the reactors was maintained at 33 ± 2 °C.

Line No. 152-154 (Revised MS):
All inorganic chemicals (purity 99%) used in the experiments are purchased from Damao (Damao Chemie Co., Ltd, Tianjin, China).

3) Authors report they used essential trace elements. What are they? Would they be found in natural conditions or would need to be supplemented in real wastewater?

Author Response: According to the van de Graaf (1996), the essential trace elements are necessary for maintaining the ANAMMOX activity. In the revised MS, we have added the information related this issue.

Line No. 150-152 (Revised MS):
The wastewater also contained trace element solution I (EDTA and FeSO4) and II (EDTA, ZnSO4·7H2O, CoCl2·6H2O, MnCl2·4H2O, CuSO4·5H2O, NaMoO4·2H2O, NiCl2·6H2O, NaSeO4·H2O and H3BO4) for ANAMMOX bacteria activity [30].

4) Also, they need to provide an explanation on why the two concentration of HS (1 and 10 mg/L) were selected.

Author Response: In preliminary research, we indeed used 0, 1, 10 and 20 mg/L of HS to study the short-term effect of HS on the nitrogen removal of synthetic wastewater (Fig. S3). In the revised MS, we have added the information related this issue.

Line No. 163-167 (Revised MS):
In preliminary research (Figure S2), we used 0, 1, 10 and 20 mg·L-1 of HS to study the short-term effect of HS on the nitrogen removal of synthetic wastewater. For the short-term effects of 10 and 20 mg·L-1 of HS on nitrogen performance are almost the equally significant, 10 mg·L-1 of HS was selected as the target concentration to study the long-term effects on nitrogen performance.

Figure S2 The short-term effects of HS on the nitrogen removal for synthetic wastewater. Error bars represent standard deviations of triplicate testes.

5) I believe this section would be further improved if authors would provide a schematic representation of the reactors design.

Author Response: As the reviewer’s suggestion, a schematic representation of the reactors design has been added in this revised manuscript (Line No. 100).

Revised MS:

Figure S1 Schematic diagram of the lab-scale anammox bioreactors. HS0, HS1 and HS10 indicated 0, 1 and 10 mg·L-1 humic substance, respectively.

# Enrichment of biological functions
Line 271: define what is meant by “satisfying” removal performance.

Author Response: The “satisfying removal performance” means “high nitrogen removal rates”. The sentence has been edited in the revised version.

Line No. 349-351 (Revised MS):
This could explain that even at decreased bacterial abundance, the ANAMMOX bioreactor could still maintain high nitrogen removal rates.

# Conclusions
Authors need to expand this section by providing suggestions and advice, based on their findings, on future research efforts on the topic.

Author Response: As the reviewer advised, conclusion sections (Line No. 462-465) have been edited in the revised manuscript.

Line No. 462-465 (Revised MS):
In the present study, the reactor could maintain satisfying performance on removing nitrogen species after long-term adaption to the low level HS (1 mg·L-1). This study demonstrated that the presence of low level HS benefited the ANAMMOX process. It can be expect that the presence of low HS potentially benefited the nitrogen removal efficiency of anammox in the wastewater treatment.

Reviewer 2 Report

The manuscript studied the impact of humic substances (HS) on ANAMMOX consortia. While it presented no significant difference in nitrogen removal by bioreactors with or without the addition of humic substances, changes in microbial structure and functional profile, particularly genes associated with nitrogen removal, were observed.

In overall, the studying topic is very interesting. The combination of metagenomics method with monitoring of operational performance is a very good approach. However, the experimental design and hence reported results does not show a relevant contribution to the scientific knowledge in this field. 

For example, HS influence is the main topic in this paper, however, the experimental design does not mention: how, what and why the HS stock was selected? Why only 1 mg/L and 10 mg/L dosages were employed?

Line 159-160, what did they define the appropriate dosage of organic matter? How can your result be compared to theirs?

With this experimental design, I cannot see the significance of the result that Authors mentioned in the abstract (line 25 – 27): “Overall, this study shed light on the important roles of a typical bio-refractory organic pool ….”

Other comments:

1)      Section 2.3: how much data of metagenomic sequencing were obtained? What is the coverage of the data? How is “high-quality reads” defined here? How is the taxonomic profile were obtained and classified?

2)      Line 327-328, with relative abundance (number of reads assigned to specific genes/total reads), with 0.101% vs. 0.076%, can it be stated “relative abundance of hzs was much higher in HS1 ….)? Similarly, in Line 348, 0.03% vs 0.043%. Or in Line 350, 0.00008% & 0.0001%, how many reads were in these percentage and how reliable it is?

3)      Line 104-105: “BOD5/COD…indicating that the …” references are required.

4)      Some p-values were included, however, I do not see what information it provided here, such as line 155, line 168. How many measurements?

5)      What is heme content? How does it tell about ANAMMOX activity?

6)      Line 168, why the effluent pH is increased?

7)      Line 332, which percentage belongs to which sample?

8)      The paper needs to be more concise. The English writing of the paper is very poor, such as line 23 (abstract), line 94-95, line 107 “harvest” lack of “ed”, line 210 “hereinafter”?, what does the term “denitrobacteria” means? line 268, “thus, we conjecture…”, etc.

Author Response

pls see the attache file.

Round 2

Reviewer 1 Report

Manuscripts present substantial improvement and addressed the majority of comments. However, while I know and understand the choice of synthetic wastewater and the advantages it brings in terms of design and interpretation of results (no need to underline text), it is important to consider that such simple conditions will never happen in a real wastewater plant and authors should explain how their main findings could then be further used for actual applications.

As they mentioned " real wastewater contains complex substances and variable compositions", so what could be the consequences of these factors on the overall reactor performance in a real situation? They have the results from their ideal, controlled situation, and on the basis of this I believe authors have all the ways to make an educated guess about what could happen or what will be needed to take into account when dealing with real wastewater.

Author Response

RESPONSE TO REVIEWERS COMMENTS (Round 2)

Re: water-426020 (Metagenomics response of ANAMMOX bacteria to bio-refractory humic substances in wastewater)

Many thanks for the constructive comments from you and the reviewers, which have helped us to further improve the quality and clarity of the manuscript.

We appreciate the opportunity to revise this manuscript, and have carefully evaluated and addressed all comments and amended the manuscript accordingly.

Authors response: Below are the itemized responses from authors to the comments of editor/reviewers (BLACK – Comments; GREEN – Authors response; RED – Revised text).

Reviewer #1 (Comments for the Author):

    Manuscripts present substantial improvement and addressed the majority of comments. However, while I know and understand the choice of synthetic wastewater and the advantages it brings in terms of design and interpretation of results (no need to underline text), it is important to consider that such simple conditions will never happen in a real wastewater plant and authors should explain how their main findings could then be further used for actual applications.

As they mentioned " real wastewater contains complex substances and variable compositions", so what could be the consequences of these factors on the overall reactor performance in a real situation? They have the results from their ideal, controlled situation, and on the basis of this I believe authors have all the ways to make an educated guess about what could happen or what will be needed to take into account when dealing with real wastewater.

Author Response: Thanks for the comments. In order to exhibited the importance of the humic substances on the overall reactor performance in a real situation, we added section 4 “Implication of this work” at the end of the revised manuscript (Line No. 505-524).

Line No. 505-524 (Revised MS):
3.6. Implication of this work

ANAMMOX process has been proved to be an effective way for nitrogen removal from landfill leachate which contains high-strength ammonium and complex organic substances [75-76]. However, it is common that organic matters become refractory with the aging of the landfills [77]. Organic nitrogen was thought to be related to humic substances (i.e., HS) and showed more recalcitrant for biodegradation [78, 79].

Previous studies were mainly focused on the roles of either microbial community or the microbial interactions in determining the anammox performance, yet molecular-level characterization of microbial interactions in the anammox performance under HS conditions have not been paid sufficient attention. In this study, therefore, the influence of HS on biological processes and the microbial community was discussed. Our current findings demonstrated that using the ANAMMOX process for treating HS-contained wastewater showed stable nitrogen removal. However, high level HS (10 mg·L-1) led to a decrease of the abundance of dominant phylum Planctomycetes and genus Candidatus Kuenenia as well as the functional genes. This phenomenon was most likely attributed to their intrinsic adaptation strategies. Thus, it can be expected that the presence of low HS potentially benefited the efficiency of ANAMMOX in the wastewater treatment. In addition, future research is needed to explore the response of bacteria activities to HS based on transcriptomics and metabolomics. Overall, our results can not only improve our knowledge of the influence of HS on anammox systems, but are also expected to supply useful information for optimizing the operation of ANAMMOX-related systems.

Reviewer 2 Report

- Authors should check the English writing more carefully to polish the manuscript. 

- In term of taxonomic classification, extracting the marker gene (i.e 16S) from metagenomic data for classification is necessary. 

Author Response

RESPONSE TO REVIEWERS COMMENTS (Round 2)

Re: water-426020 (Metagenomics response of ANAMMOX bacteria to bio-refractory humic substances in wastewater)

Many thanks for the constructive comments from you and the reviewers, which have helped us to further improve the quality and clarity of the manuscript.

We appreciate the opportunity to revise this manuscript, and have carefully evaluated and addressed all comments and amended the manuscript accordingly.

Authors response: Below are the itemized responses from authors to the comments of editor/reviewers (BLACK – Comments; GREEN – Authors response; RED – Revised text).

Reviewer #2 (Comments for the Author):

1) Authors should check the English writing more carefully to polish the manuscript.

Author Response: In the revised MS, we checked the English writing more carefully.

2) In term of taxonomic classification, extracting the marker gene (i.e 16S) from metagenomic data for classification is necessary.

Author Response: In the revised MS, marker gene from metagenomic data have been added (Line No. 220-222 and Line No. 277-280).

Line No. 220-222 (Revised MS):
Furthermore, to extract reads matching the ribosomal small subunit for taxonomic classification, the SILVA SSU RefNR99 dataset (version 115) was used as reference [41].

Line No. 277-280 (Revised MS):
The composition of the microbiome was also analysed using 16S rRNA gene reads retrieved from the metagenomes (Figure S3), showing that HS could decrease the abundance of dominant phylum Planctomycetes (43.21%, 37.27% and 37.42% in HS0, HS1 and HS10, respectively).

Figure S3 The community's taxonomic composition calculated from metagenomic data by extracting the marker gene for classification. HS0, HS1 and HS10 indicated 0, 1 and 10 mg·L-1 humic substance, respectively.
